# Stability and Heat Input Controllability of Two Different Modulations for Double-Pulse MIG Welding

**Jiaxiang Xue \*, Min Xu, Wenjin Huang, Zhanhui Zhang** **, Wei Wu and Li Jin**

School of Mechanical and Automotive Engineering, South China University of Technology, Guangzhou 510641, China; scut221@126.com (M.X.); yunji768@126.com (W.H.); 201610100399@mail.scut.edu.cn (Z.Z.); 201710100398@mail.scut.edu.cn (W.W.); jinli8756@163.com (L.J.)

**\*** Correspondence: mejiaxue@scut.edu.cn; Tel.: +86-020-2223-6360

**Abstract:** Aluminum alloy welding frequently experiences difficulties such as heat input control, poor weld formation, and susceptibility to pore generation. We compared the use of two different modulations for double-pulse metal inert gas (MIG) welding to reduce the heat input required to generate oscillations in the weld pool. The stabilities of rectangular wave-modulated and trapezoidal wave-modulated double-pulse MIG welding (DP-MIG and TP-MIG) were analyzed by examining their welding processes and weld profiles. We found that the transitional pulse in TP-MIG welding results in smoother current transitions, softer welding arc sounds, and a highly uniform fish-scale pattern. Therefore, TP-MIG welding is more stable than DP-MIG welding. The effects of these double-pulse modulation schemes on welding input energy are presented. We propose methods for reducing welding input energy by varying the number of pulses or the pulse base time of low-energy pulse train while keeping the welding current and welding arc stable and unchanged. Compared to DP-MIG welding, TP-MIG welding reduces the input energy by 12% and produces finer grain sizes, which increases weld hardness. Therefore, TP-MIG welding offers a new approach for heat input control in DP-MIG welding of aluminum alloys. The results of this work are significant for aluminum alloy welding.

**Keywords:** double-pulse MIG welding; rectangular waves; trapezoidal waves

## 1. Introduction

Despite the constant evolution of welding materials and structures, aluminum is still considered to be an extremely promising welding material owing to its abundance, lightness, corrosion resistance, and ductility [1]. Approximately 91% of all industries in China use aluminum products, including container production, extrusion profile manufacturing, rail vehicle manufacturing, and aerospace. Therefore, the development of the aluminum industry is closely related to GDP growth in China [2,3]. Today, the rapid development of the economy and intense market competition has led to a constant requirement for a higher level of productivity. Consequently, the developments in welding technology have been dominated by increases in welding speed [4]. High-speed gas metal arc welding is performed at speeds higher than 1 m/min [5]. However, performing high-speed welding using aluminum materials is challenging, owing to the physical properties of aluminum. For example, the electrical conductivity of aluminum is 62% IACS (International Annealed Copper Standard), which is approximately 6 times that of low-carbon steel, whereas its thermal conductivity is 222 [W/(m·K)] at 25 °C, which is approximately 5 times that of low-carbon steel, making high-speed welding of aluminum more difficult [6–9]. Nonetheless, the continuous development of welding techniques has been accompanied by improvements in high-speed aluminum welding techniques.

The welding speed and welding current can affect heat input, which can also affect the microstructure and mechanical properties [10–13]. Pulsed gas metal arc welding (P-GMAW) is a method based on electric pulses that was invented in the 1960s by English researchers. In this method, the melting of the welding wire and base metal is controlled by peak pulse current, whereas the pulse base current maintains the welding arc. This technique has significantly improved the range of adjustment for welding current. In addition, this technique prevents weld orientation from having a significant impact on welding processes during droplet transfer. However, the welds formed using the P-GMAW technique are frequently accompanied by pores, which impairs the finish and thus limits the applicability of this technique [14].

Double-pulse GMAW (DP-GMAW) is based on the P-GMAW technique; it involves the modulation of high-frequency pulses by a low-frequency waveform [15]. The advantages of DP-GMAW are lower costs and higher suitability for outdoor applications. The DP-GMAW technique has various implementations. One approach is the low-frequency switching of wire feed speed and welding current (e.g., the Kemppi Pro Evolution) [16], while another approach is the low-frequency switching of the melting rate at a constant wire feed speed (e.g., OTC welding systems). At present, the low-frequency modulation of welding current provides a better level of welding quality [17]. In this mode of modulation, low frequencies are used to modulate two large-amplitude high-frequency pulse trains. High-frequency current facilitates droplet transfer, whereas low-frequency modulation oscillates the weld pool using strong and weak pulse trains. This helps in rapidly ejecting hydrogen gas from the weld pool and thus reduces the formation of pores in the weld [15].

In the design of current pulse parameters for double-pulse welding, different transfer modes can be used to switch the currents of strong and weak pulses. In this work, we investigated the welding stability and heat input controllability of two methods for the switching of strong and weak pulses, namely, rectangular-pulse metal inert gas (DP-MIG) welding and trapezoidal-pulse MIG (TP-MIG) welding. The findings of this work will be useful for the design of process parameters in double-pulse MIG welding.

## 2. Design of Current Waveform Parameters for Double-Pulse MIG Welding with Two Different Modulations

In double-pulse MIG welding, waveform control refers to the modulation of high-frequency pulses using a low-frequency rectangular waveform. Strong and weak pulse trains are continuously alternated during this process, which forces gases formed by the welding processes out of the weld. Therefore, double-pulse MIG welding reduces the effects of weld pores, reduces the sensitivity of the joint to cracks, and improves the finish and mechanical properties of the weld [18].

Pulsed MIG welding is an important breakthrough in welding technology, as it has eliminated numerous constraints of constant-current welding techniques. Moreover, the advent of this technology has provided a new direction for research and development in the field of arc welding [19]. The energy used by pulsed MIG welding can be controlled by adjusting time and current parameters. This ensures that low current will be used during the welding of aluminum alloy plates, in accordance with the requirements of aluminum alloy processing [20]. The relationship between current and the transfer mode of double-pulsed MIG welding is illustrated in Figure 1.

The globular transfer mode is essentially open-arc welding, and it has lower levels of current compared to spray transfer. As shown in Figure 2, the welding current of the spray transfer mode increases in proportion with the surface tension of molten metal, and this mode uses higher currents than several other modes of transfer. This mode of transfer is suitable for settings with low base voltages and high peak currents [21]. The streaming transfer mode leads to higher currents, lower droplet volumes, and formation of axial droplet columns. If the current exceeds 300 A, the electrical resistivity of the transfer process increases. The control of current in pulsed transfer is performed by switching between globular transfer and spray transfer [22].

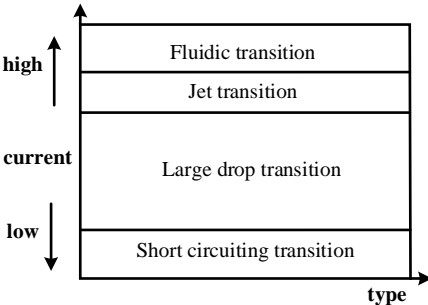

**Figure 1.** Relationship between transfer mode and current.

Figure 2 illustrates the current waveform of single-pulse MIG (P-MIG) welding. This process only involves a few factors, and the most important parameters are the peak current of the pulse, $I_p$, and the base current of the pulse, $I_b$. It should be noted that $I_p$ and $I_b$ are the extreme values that manifest during the adjustment period of the pulses. As $I_b$ is typically very small, it is sufficient only for powering idle operation and maintaining the welding arc. $I_b$ also plays a role in the preheating process.

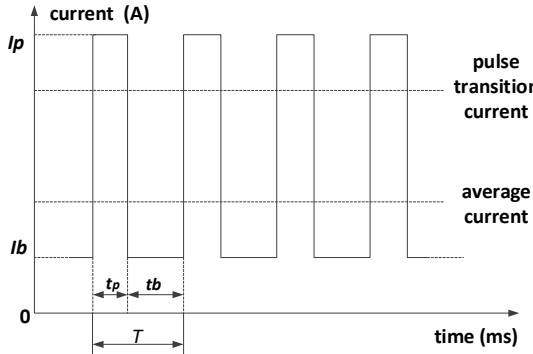

**Figure 2.** Sketch of the P-MIG scheme.

Double-pulse MIG welding effectively decreases crack sensitivity and pore generation rates. In addition, the welding patterns formed by this method are aesthetically pleasing. As shown in Figure 3, low-energy and high-energy pulse trains alternate with each other in double-pulse MIG welding. The meanings of the parameters shown in Figure 3 are as follows: $I_{bw}$ ($I_{pw}$) and $t_{bw}$ ($t_{pw}$) represent the base (peak) current and base (peak) time of the pulses in the low-energy pulse train, respectively; $I_{bs}$ ($I_{ps}$) and $t_{bs}$ ($t_{ps}$) represent the base (peak) current and base (peak) time of the high-energy pulse train, respectively; $T_w$ is the period of the low-energy pulse train; and n and m represent the number of high energy pulses and low-energy pulses per unit welding current cycle, respectively.

The alternation of high-energy and low-energy pulse trains over the duration of $T$ (i.e., the time required to complete the processes of double-pulse MIG welding) causes the welder to alternate between hot and cold states [23]. As aluminum alloys have a low melting point and high conductivity, these materials are easily affected by temperature, which could lead to hot cracks and weld damage. The possibility of oxide film breakdown and the weakening of base material strength in the presence of high energy levels make it necessary to strictly control the input of welding energy.

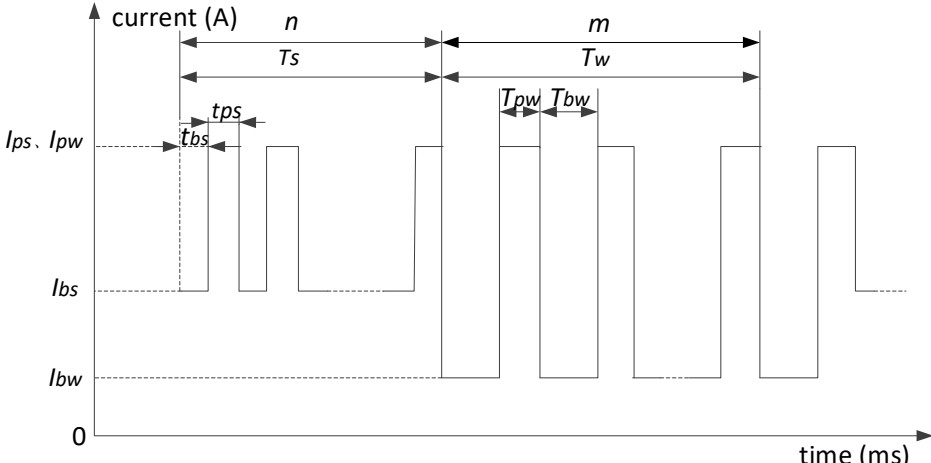

**Figure 3.** DP-MIG current waveform diagram.

Owing to welding behaviors and environmental effects, the arc length during double-pulse MIG welding is always in a varying state. Thus, double-pulse MIG welding current has a rectangular waveform. When the pulse train undergoes a transition from high energy to low energy, the arc length and wire extension continue to increase until they reach their maximum values, whereas the current parameter decreases. This results in considerable change in energy. The reduced current eventually becomes insufficient to sustain the arc, which causes the arc to breakdown. The effects of this type of damage on the base metal are particularly significant in low-temperature environments. In this work, trapezoidal waveforms were investigated to address this issue. Our new approach is referred to as TP-MIG welding; the current waveform of this scheme is shown in Figure 4.

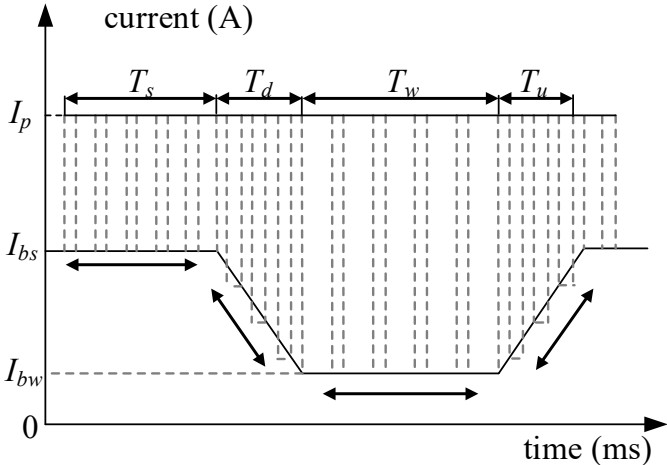

**Figure 4.** TP-MIG low-frequency modulation waveform.

In TP-MIG welding, the transition of a high-energy pulse train to a low-energy pulse train in DP-MIG welding is supplemented by a transitional high- to low-energy pulse train, whereas the opposite transition (low-energy pulse train to high-energy pulse train) is supplemented by a transitional low- to high-energy pulse train. The rectangular waveform exhibits trapezoidal characteristics owing to the transitional pulse trains. When the arc extends to its maximum length during the high-energy to low-energy transition of the pulse train, the current gradually decreases because of the pulse train, which is conducive for controlling arc breakdown. This subsequently improves the efficacy of aluminum alloy welding and ensures the construction of an energy gradient that improves the controllability and stability of the welding process.

The analysis of low-frequency modulation signals includes two aspects: peak values and base values. If the peak values are stable, low-frequency modulation may be performed using $I_p$. The change in base current along the four aforementioned modes of transfer is shown in Figure 4. $T_d$ and $T_u$ are the low- to high-energy and high- to low-energy pulse transition times, respectively, and $T_d = T_u$. In Figure 4, $T_l$ is the time required to complete the processes of low-frequency modulation, and the corresponding frequency is $f_l = \frac{1}{T_l}$. Hence, the time, $t$, at which the interval $[0, T_l]$ lies in the low-frequency period is as follows:

$$\widetilde{t}_1 = t - INT\left(\frac{t}{T}\right) \times T_1. \tag{1}$$

Therefore, the base current can be expressed as follows for all intervals of the low-frequency signal:

$$I_{bl} = \begin{cases} I_{bs} & 0 \leq \widetilde{t}_l \prec T_s \\ I_{bs} - \dfrac{I_{bs} - I_{bw}}{T_d}(\widetilde{t}_l - T_s) & T_s \leq \widetilde{t}_l \prec T_s + T_d \\ I_{bw} & T_s + T_d \leq \widetilde{t}_l \prec T_s + T_d + T_w \\ I_{bw} + \dfrac{I_{bs} - I_{bw}}{T_d}(\widetilde{t}_l - T_s - T_d - T_w) & T_s + T_d + T_w \leq \widetilde{t}_l \prec T \end{cases} \tag{2}$$

The analysis of high-frequency modulation signals involves four different stages. The first stage is the high-energy pulse train. In this stage, the energy is highly concentrated, and the melting depth of the weld pool is the most important regulatory factor. The next stage is the transitional high- to low-energy pulse, followed by the low-energy pulse train. The behaviors in this stage include preheating of the material and stabilization of the weld pool. The final stage is the transitional low- to high-energy pulse. During this stage, the peak currents of the high-energy and low-energy pulse trains are always the same, but there are marked differences in their base currents (Figure 5).

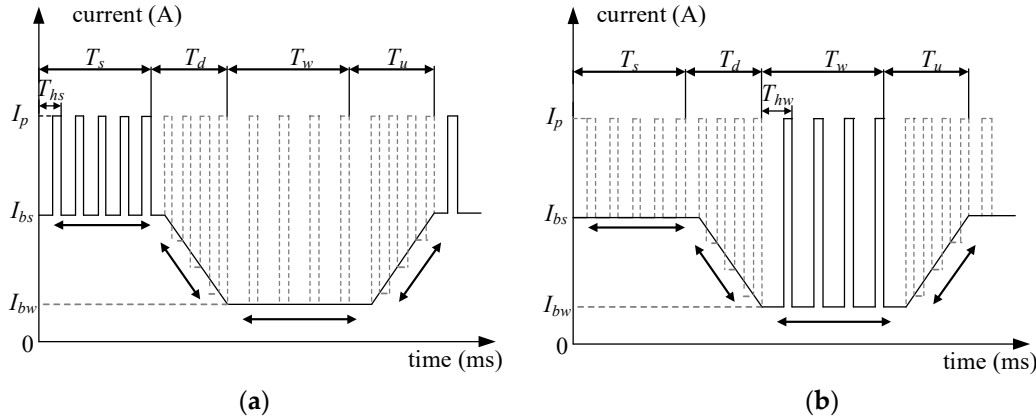

**Figure 5.** High-frequency modulation signal energy pulse group graph: (**a**) High-energy pulse train of the high-frequency modulation signal; (**b**) Low-energy pulse train of the high-frequency modulation signal.

Suppose that high-frequency modulation occurs over a period of $T_h$, and the period of the high-energy pulse train is $T_{hs}$, corresponding to a frequency of $f_{hs} = 1/T_{hs}$. The peak of the high-energy pulse train is the same as that of the low-frequency modulation signal, whereas its base value is $I_{bs}$. The duty cycle is $D_s$. Based on these parameters, the corresponding time of the high-frequency signal in $[0, T_{hs}]$ is:

$$\widetilde{t}_{hs} = \widetilde{t}_l - INT\left(\frac{\widetilde{t}_l}{T_{hs}}\right) \times T_{hs}. \tag{3}$$

The current values corresponding to this period can be expressed as:

$$I_{hs} = \left(\frac{I_p + I_{bs}}{2}\right) - \left(\frac{I_p - I_{bs}}{2}\right) sign\left\{\widetilde{t}_l - INT\left(\frac{\widetilde{t}_l}{T_{hs}}\right) \times T_{hs} - D_s T_{hs}\right\}. \tag{4}$$

According to the same principles, the period of the low-energy pulse train is $T_{hw}$, and its interval is $[T_s + T_d, T_s + T_d + T_{hw}]$. In this interval, the time and current values corresponding to the high-frequency signal are:

$$\widetilde{t}_{hw} = \widetilde{t}_l - T_s - T_d - INT\left(\frac{\widetilde{t}_l - T_s - T_d}{T_{hw}}\right) \times T_{hw} \tag{5}$$

$$I_{hs} = \left(\frac{I_p + I_{bs}}{2}\right) - \left(\frac{I_p - I_{bs}}{2}\right) sign\left\{\widetilde{t}_l - T_s - T_d - INT\left(\frac{\widetilde{t}_l - T_s - T_d}{T_{hs}}\right) \times T_{hs} - D_s T_{hw}\right\}. \tag{6}$$

It should be noted that the following relation must be satisfied to ensure the transfer of one droplet for each pulse:

$$D_s T_{hs} = D_w T_{hw}. \tag{7}$$

This can be rewritten as:

$$\frac{D_s}{D_w} = \frac{T_{hw}}{T_{hs}} = \frac{f_{hs}}{f_{hw}}. \tag{8}$$

The high- to low-energy and low- to high-energy transitions have a fixed time span, and the corresponding energy variation plots are shown in Figure 6. In the interval $[T_s, T_s + T_d]$ of the high- to low-energy transition, the base current can be expressed as:

$$I_b = I_{bs} - \frac{I_{bs} - I_{bw}}{T_d}(\widetilde{t}_1 - T_s). \tag{9}$$

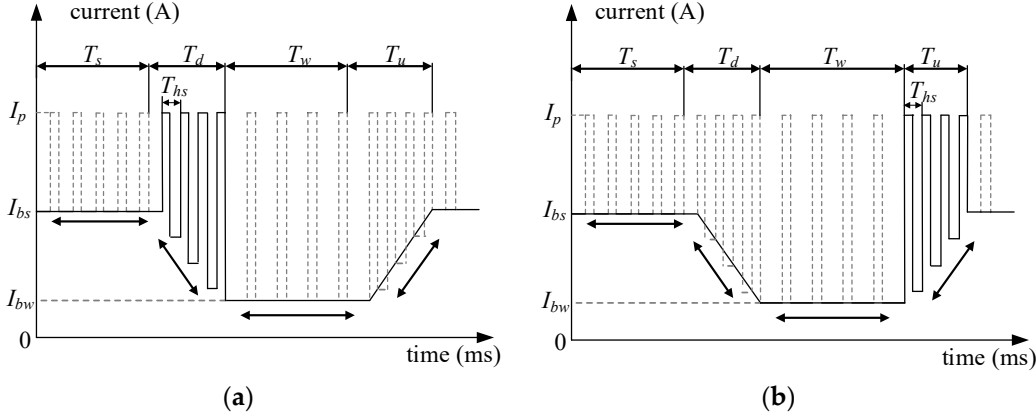

**Figure 6.** Transition pulse group of high-frequency modulation signal: (**a**) High- and low-energy transition pulse group; (**b**) Low-energy transition pulse group.

The period in Figure 6a that corresponds to the interval $[T_s, T_s + T_{hs}]$ is:

$$\widetilde{t}_{hd} = \widetilde{t}_l - INT\left(\frac{\widetilde{t}_l - T_s}{T_{hs}}\right) \times T_{hd} - T_s. \tag{10}$$

The corresponding currents are given as:

$$I_{hd} = \left(\frac{I_p + I_{bs} - \frac{I_{bs} - I_{bw}}{T_d}(\widetilde{t}_1 - T_s)}{2}\right) - \left(\frac{I_p - I_{bs} + \frac{I_{bs} - I_{bw}}{T_d}(\widetilde{t}_1 - T_s)}{2}\right) sign\left\{\widetilde{t} - INT\left(\frac{\widetilde{t}_1 - T_s}{T_{hs}}\right) \times T_{hs} - T_s - D_s T_{hs}\right\}. \tag{11}$$

Similarly, the base current corresponding to the interval $[T_s + T_d + T_w, T]$ of the transitional low-to high-energy pulse train can be determined. In Figure 6b, the period corresponding to the interval $[T_s + T_d + T_w, T]$ is:

$$\widetilde{t}_{hl} = \widetilde{t}_l - INT\left(\frac{\widetilde{t}_l - T_s - T_d - T_w}{T_{hs}}\right) \times T_{hd} - T_s - T_d - T_w. \tag{12}$$

The corresponding current values can be expressed as:

$$I_{hl} = \left(\frac{I_p + I_{bw} + \dfrac{I_{bs} - I_{bw}}{T_d}(\widetilde{t}_1 - T_s - T_d - T_w)}{2}\right) - \left(\frac{I_p - I_{bw} - \dfrac{I_{bs} - I_{bw}}{T_d}(\widetilde{t}_1 - T_s - T_d - T_w)}{2}\right)$$
$$\times sign\left\{\widetilde{t}_1 - INT\left(\frac{\widetilde{t}_1 - T_s - T_d - T_w}{T_{hs}}\right) \times T_{hs} - T_s - D_s T_{hs}\right\} \tag{13}$$

The calculations show that the total number of high- (low-) energy pulse trains will decrease with increases in $t_{bw}$ and $t_{bs}$, if $T_w$ and $T_s$ are fixed. Therefore, if the current is to be limited to lower values, it is necessary to reduce $n$ and $m$ while appropriately increasing $t_{bw}$ and $t_{bs}$. In addition, the dependence of current on the number of pulses must be closely monitored to minimize the latter's impact on current.

In conventional DP-MIG welding, the energy of the welding processes undergoes rapid switching, which is detrimental to welding stability. In addition, this increases the probability of screw-like wires being formed when the arc is started in "cold plate" conditions.

## 3. Comparison of Stabilities of DP-MIG Welding and TP-MIG Welding

The stabilities of TP-MIG welding and DP-MIG welding were compared by analyzing the sound of their arcs, their transient welding current waveforms, and their weld morphologies.

An experiment was conducted using an automated welding system and a 1.2-mm (ER4043) aluminum alloy welding wire. The weld was formed on an AA6061 alloy plate (with thicknesses in the range of 1–5 mm) at a constant wire feed speed. The standardized current levels in this experiment were 60 A, 80 A, 100 A, 120 A, 140 A, 160 A, 180 A, and 200 A. The DP-GMAW long-step calibration curve for 1.2 mm ER4043 aluminum alloy welding wires (with $t_{bs}$ and $t_{bw}$ being 8 ms, $n_s$ and $n_w$ being 15, $I_{ps}$ being 470 A, $I_{pw}$ being 330 A, and $I_{bs}$ and $I_{pw}$ being 45 A) was constructed using expert data is shown in Figure 7.

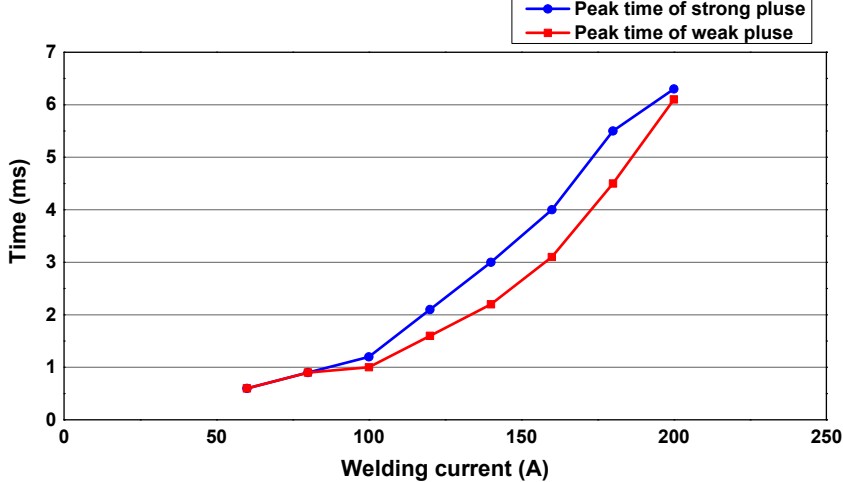

**Figure 7.** 1.2-mm calibration curve step distance of aluminum alloy welding wire.

### 3.1. Process Stability of DP-MIG Welding

The current waveform and welding morphology corresponding to a welding current of 80 A are shown in Figure 8. The figure shows that DP-GMAW creates fish-scale patterns that are more aesthetically pleasing, and sharper welding-arc sounds. Welding was performed using numerical values generated by the Newton's interpolation algorithm. The results of this experiment are shown in Figure 9. The figure shows that fish-scale patterns of higher quality were produced using the numerical values generated by the Newton–Raphson method in welding operations. This result highlights the superiority of computational algorithms in autoregulation of welding operations.

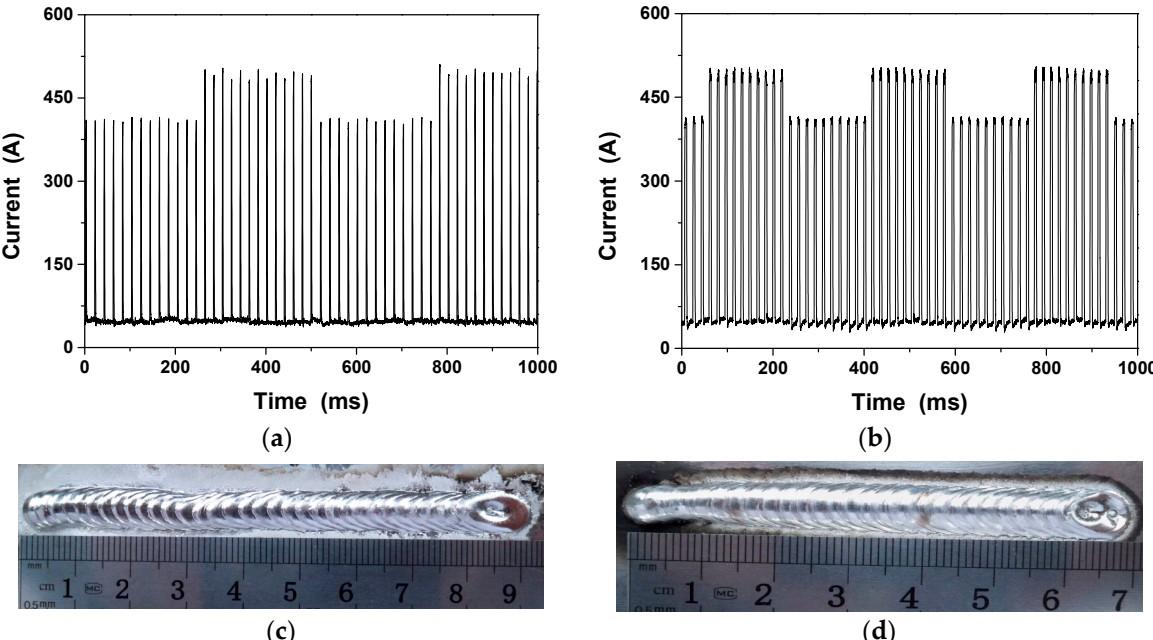

**Figure 8.** Waveform and weld formation at 80 A and 160 A: (**a**) 80 A welding waveform; (**b**) 160 A welding waveform; (**c**) 80 A weld formation; (**d**) 160 A weld formation.

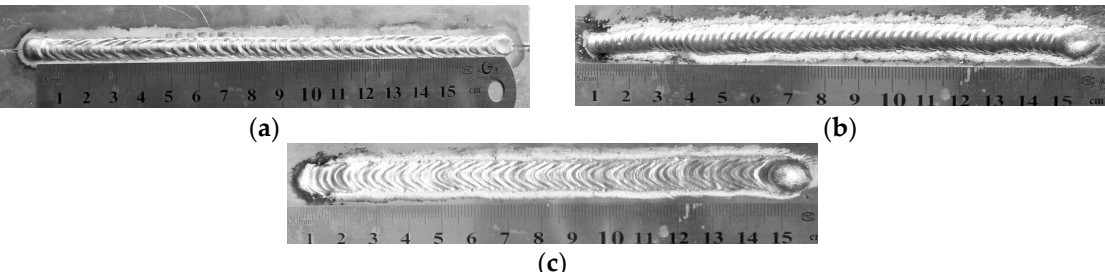

**Figure 9.** Weld seam obtained by parameter self-adjusting welding test; (**a**) 83 A weld formation; (**b**) 92 A weld formation; (**c**) 135 A weld formation.

### 3.2. Process Stability of TP-MIG Welding

A TP-MIG welding experiment was conducted using a fixed current of 120 A and a range of frequencies, and the results of this experiment are shown in Figure 10. According to the weld morphologies shown in this figure, a frequency of 2 Hz results in extremely clearly defined fish-scale patterns and large gaps between the fish scales. Increasing the frequency to 5 Hz significantly improves the weld quality and the density of the fish-scale patterns. A further increase in welding frequency to 15 Hz results in a corresponding increase in the density of the fish scales and the manifestation of single-pulse features. In addition, it is found that the transitional pulses in TP-MIG help stabilize

current transitions and reduce the sharpness of welding-arc sounds. In summary, TP-MIG welding is highly stable and produces high-quality welds that are aesthetically pleasing.

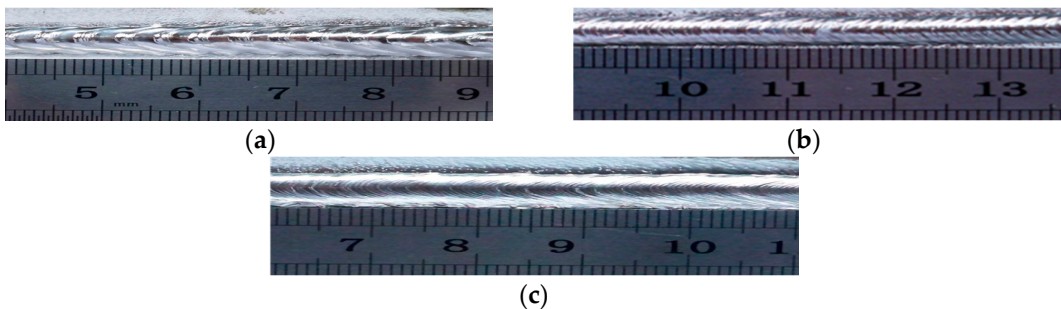

**Figure 10.** 120 A waveform and weld formation at different frequencies: (**a**) 2 Hz; (**b**) 5 Hz; (**c**) 15 Hz.

## 4. Process Control Experiment on Reduction of Heat Input in TP-MIG Welding and DP-MIG Welding

A comparative experiment was conducted using TP-MIG welding and DP-MIG welding, where the input energy decreased by varying the number of pulses in the pulse train, the frequency of the low-energy pulse train, and the welding speed. The effects of rectangular and trapezoidal modulation on internal weld quality were analyzed by comparing the resulting weld morphologies and metallurgical structures. Table 1 presents the chemical compositions of the base metals and welding wires used in the MIG welding processes.

**Table 1.** Chemical composition of AA6061 material and ER4043 wire (mass fraction%).

| Material | Mg | Si | Fe | Cu | Mn | Ti | Al |
|----------|------|------|------|----------|------|------|------|
| AA6061 | 0.80~1.2 | 0.4~0.8 | 0.7 | 0.15~0.40 | 0.15 | 0.15 | Bal. |
| ER4043 | 0.05 | 5.60 | 0.80 | 0.30 | 0.05 | – | Bal. |

### 4.1. Reducing Heat Input by Varying Base Time by Adjusting the Number of Pulses in the Low-Energy Pulse Train

The energy of the welding arc is transferred to the base metal, and part of this energy increases the temperature of the welding wire. The total energy of the welding arc is given by $P = IU$, and the effective welding power is $P_\lambda = \lambda IU$. The effective power of these energy sources is in the range of 0.70–0.80; in this work, $P_\lambda$ is considered to be 0.70. Based on the principles of DP-MIG welding and TP-MIG welding, it can be inferred that if current is to be reduced in these processes, the corresponding number of pulses should also be reduced. If the frequency is unchanged, decreasing the number of pulses will increase the base time. Based on the input energy equation, $q = P_\lambda/v$, decreasing the current will result in a proportionate decrease in the effective power, if the input frequency is fixed.

A comparative experiment was conducted using DP-MIG welding and TP-MIG welding at a welding speed of 0.6 m/min and constant current. The other parameters in this experiment are listed in Tables 2 and 3. The appearances of the welds formed by TP-MIG welding and DP-MIG welding at each level of average current are shown in Figures 11 and 12, whereas Figures 13 and 14 are the corresponding waveform plots.

**Table 2.** DP-MIG test data at different number of pulses in the low-energy pulse train.

| $I_{ps}$ (A) | $t_{ps}$ (ms) | $I_{bs}$ (A) | $t_{bs}$ (ms) | $n$ (number) | $I_{pw}$ (A) | $t_{pw}$ (ms) | $I_{bw}$ (A) | $t_{bw}$ (ms) | $m$ (number) | Average Current (A) | Average Voltage (V) | Input Energy (J) |
|------|------|------|------|------|------|------|------|------|------|------|------|------|
| 210 | 2 | 72 | 6 | 16 | 210 | 2 | 48 | 15 | 8 | 88 | 22 | 135,520 |
| 210 | 2 | 72 | 6 | 16 | 210 | 2 | 48 | 26 | 5 | 80 | 21 | 117,600 |

**Table 3.** TP-MIG test data at different number of pulses in the low-energy pulse train.

| $I_{ps}, I_{psw},$ $I_{pw}, I_{pws}$ (A) | $t_{ps}, t_{psw},$ $t_{pw}, t_{pws}$ (ms) | $I_{bs}$ (A) | $t_{bs}$ (ms) | $n$ | $I_{bw}$ (A) | $t_{bw}$ (ms) | $m$ | $n_{sw}$ | $n_{ws}$ | Average Current (A) | Average Voltage (V) | Input Energy (J) |
|---|---|---|---|---|---|---|---|---|---|---|---|---|
| 210 | 2 | 72 | 6 | 15 | 48 | 25 | 6 | 4 | 4 | 88 | 22 | 135,520 |
| 210 | 2 | 72 | 6 | 15 | 48 | 50 | 3 | 4 | 4 | 80 | 21 | 117,600 |

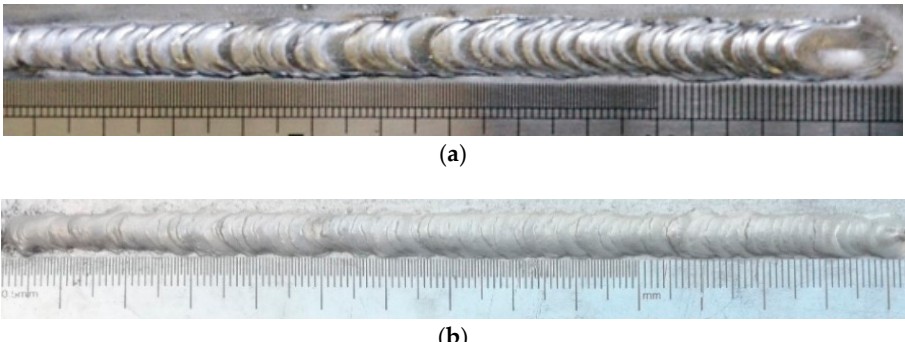

**Figure 11.** DP-MIG weld under different average currents: (**a**) Average current of 88 A; (**b**) Average current of 80 A.

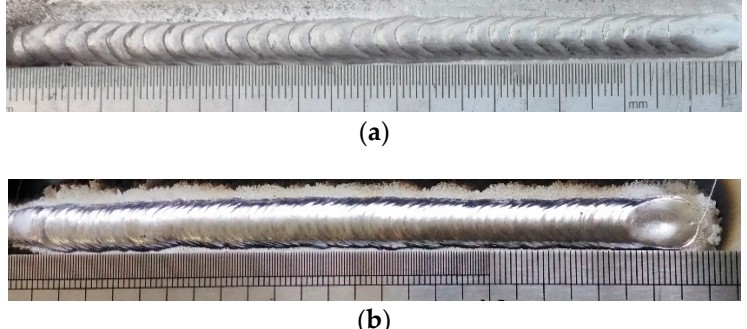

**Figure 12.** TP-MIG weld under different average currents: (**a**) Average current of 88 A; (**b**) Average current of 80 A.

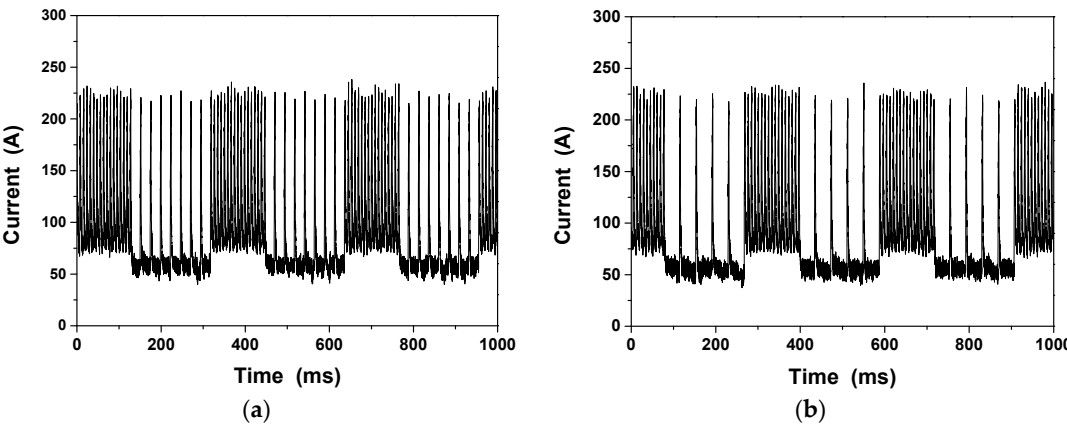

**Figure 13.** DP-MIG current waveform diagram of different number of pulses in the low-energy pulse train: (**a**) Average current of 88 A; (**b**) Average current of 80 A.

Figures 11–14 demonstrate that the average welding current can be reduced by decreasing the number of pulses in the low-energy pulse train if the parameters of the high-energy pulse train are kept unchanged. The input energy can be reduced by approximately 14% using this approach. The welds in Figures 11 and 12 are normal. However, the fish scales of the TP-MIG weld are more uniform. Micrographs of the welds produced by different currents in TP-MIG welding were compared with

those produced by DP-MIG welding, as shown in Figures 15 and 16. Grain growth is more severe in Figures 15a and 16a compared to Figures 15b and 16b, and DP-MIG welds have fewer second-phase particles. These results indicate that a high energy input induces grain growth and reduces the number of second phase particles. Therefore, reducing the number of pulses in the low-energy pulse train effectively reduces the input energy. Furthermore, a comparison of Figures 15 and 16 shows that the TP-MIG weld has finer grain sizes. This shows that the molten metals scour and shear in the presence of alternating strong and weak welding arcs, which prevents stable grain growth. Furthermore, the fragmented grains induce the appearance of heterogeneous nucleation points, which is conducive to grain refinement.

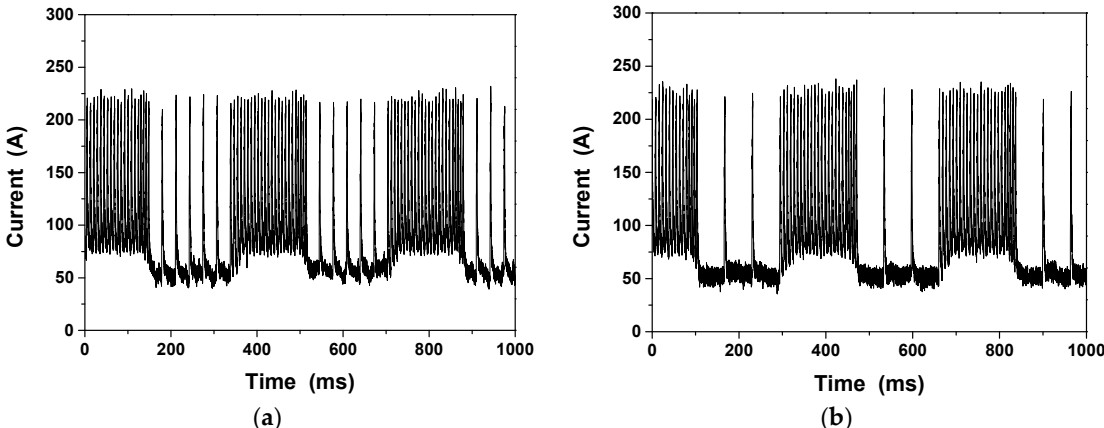

(a)                                                   (b)

**Figure 14.** TP-MIG current waveform diagram of different number of pulses in the low-energy pulse train: (**a**) Average current of 88 A; (**b**) Average current of 80 A.

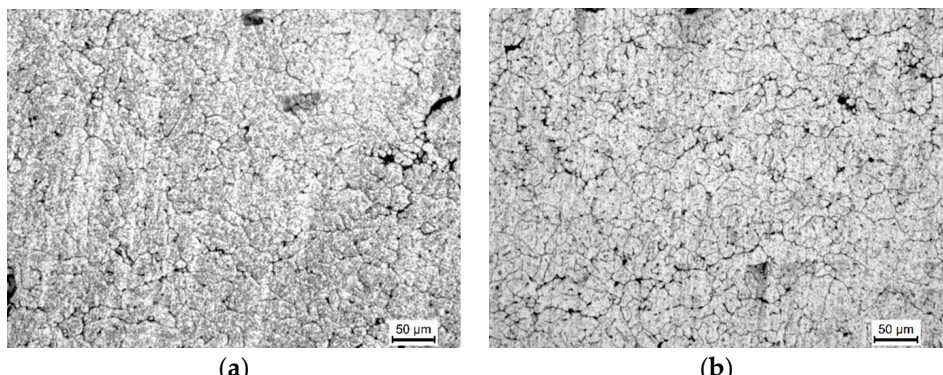

(a)                                                   (b)

**Figure 15.** Micrograph of the welds of DP-MIG under different average currents: (**a**) Average current of 88 A; (**b**) Average current of 80 A.

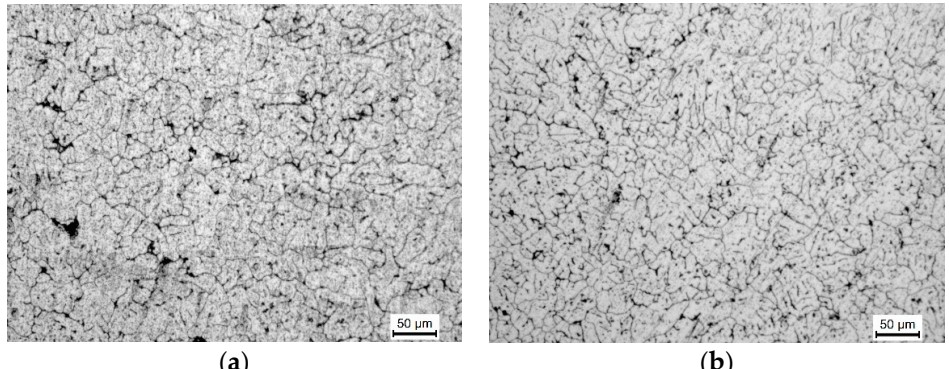

(a)                                                   (b)

**Figure 16.** Micrograph of the welds of TP-MIG under different average currents: (**a**) Average current of 88 A; (**b**) Average current of 80 A.

### 4.2. Reducing Heat Input by Adjusting the Frequency of Low-Energy Pulse Train to Vary the Base Time

If the parameters of the high-energy pulse train and the peak current and peak times of the low-energy pulse train are fixed, the heat input can be reduced by increasing the duration of the base current in the low-energy pulse train because this effectively reduces the average current. This approach is known as *variation in the base time*. Two DP-MIG welding experiments were conducted at a welding speed of 0.6 m/min, and the parameters are listed in Table 4. The appearance and morphology of the welds formed in these experiments and their corresponding current waveform plots are illustrated in Figure 17.

**Table 4.** Comparison of DP-MIG test data of low-energy pulse group at different frequencies.

| $I_{ps}$ (A) | $t_{ps}$ (ms) | $I_{bs}$ (A) | $t_{bs}$ (ms) | $n$ | $I_{pw}$ (A) | $t_{pw}$ (ms) | $I_{bw}$ (A) | $t_{bw}$ (ms) | $m$ | Average Current (A) | Average Voltage (V) | Input Energy (J) |
|---|---|---|---|---|---|---|---|---|---|---|---|---|
| 210 | 2 | 77 | 6 | 16 | 210 | 2 | 50 | 13 | 9 | 90 | 22 | 138,600 |
| 210 | 2 | 77 | 6 | 16 | 210 | 2 | 50 | 22 | 9 | 80 | 22 | 123,200 |

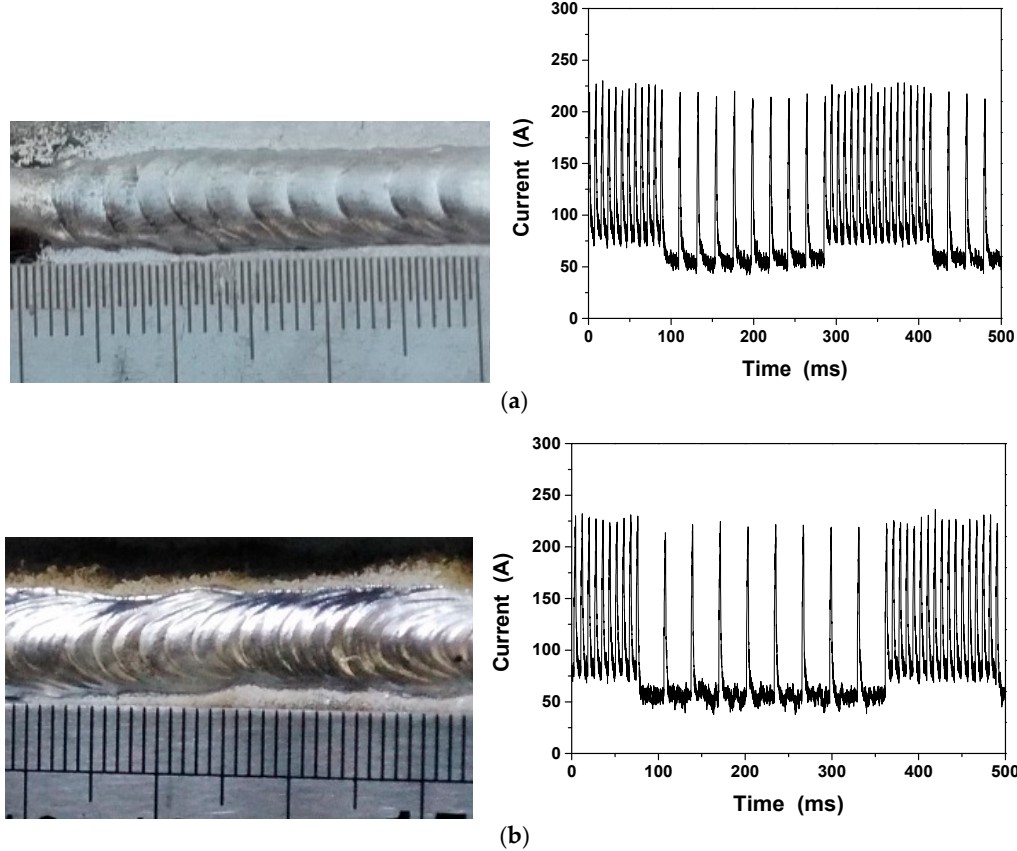

**Figure 17.** DP-MIGW at different frequencies: (**a**) Average current of 90 A, frequency of 3.8 Hz; (**b**) Average current of 80 A, frequency of 3 Hz.

Two TP-MIG welding experiments were also conducted at the same welding speed (0.6 m/min), and the parameters are listed in Table 5. The superficial morphologies of the welds formed in these experiments and their corresponding current waveforms are shown in Figure 18.

**Table 5.** Comparison of TP-MIGW test data of low-energy pulse group at different frequencies.

| $I_{ps}, I_{psw}, I_{pw}, I_{pws}$ (A) | $t_{ps}, t_{psw}, t_{pw}, t_{pws}$ (ms) | $I_{bs}$ (A) | $t_{bs}$ (ms) | $n$ | $I_{bw}$ (A) | $t_{bw}$ (ms) | $m$ | $n_{sw}$ | $n_{ws}$ | Average Current (A) | Average Voltage (V) | Input Energy (J) |
|---|---|---|---|---|---|---|---|---|---|---|---|---|
| 210 | 2 | 77 | 6 | 15 | 50 | 20 | 7 | 4 | 4 | 90 | 22 | 138,600 |
| 210 | 2 | 77 | 6 | 15 | 50 | 30 | 7 | 4 | 4 | 80 | 22 | 123,200 |

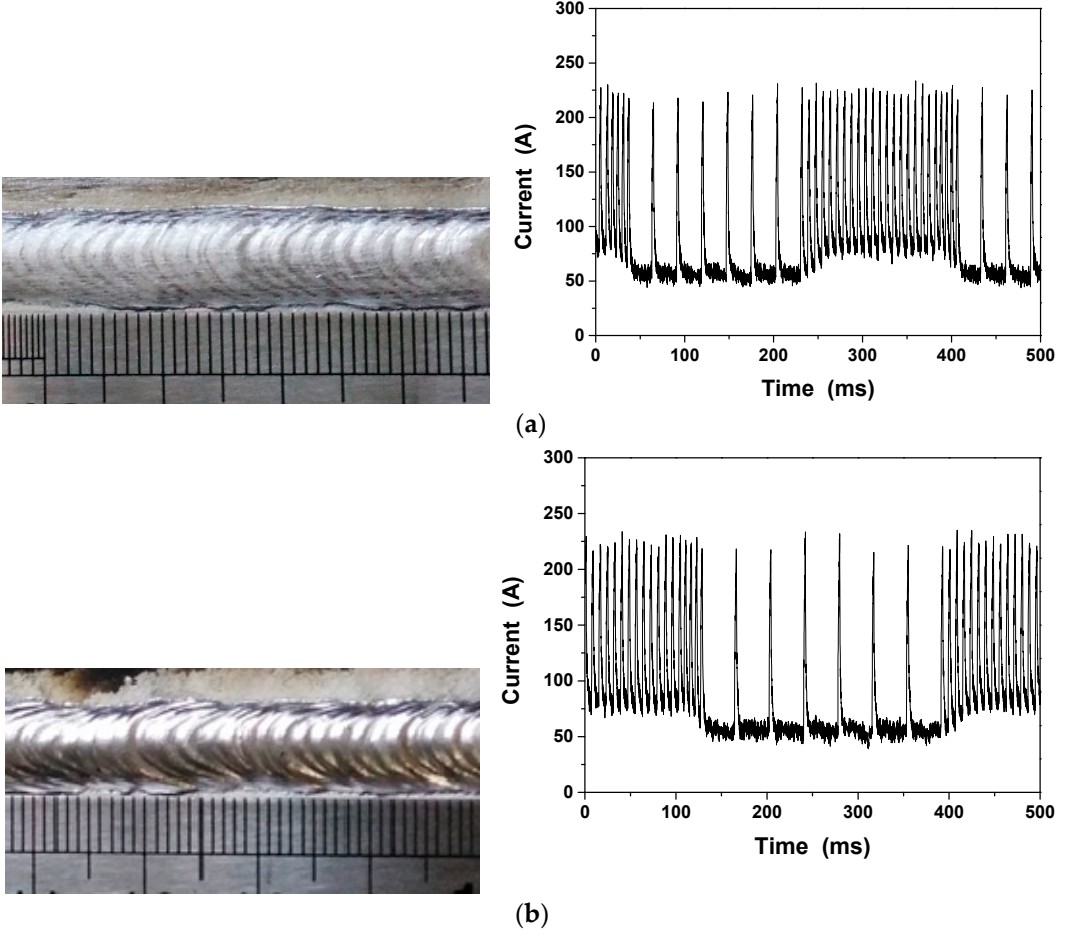

**Figure 18.** TP-MIGW at different frequencies; (**a**) Average current of 90 A, frequency of 3 Hz; (**b**) Average current of 80 A, frequency of 2.4 Hz.

This approach reduces the input energy by approximately 12%. The average current and frequency decrease as the base time of the low-energy pulse train increases, and the fish scales formed at the welded joint become denser.

### 4.3. Effects of Welding Speed on Input Energy

The welding arc generates heat, and a heat-affected zone is created close to the joint. An increase in the welding speed leads to a proportionate decrease in heat transfer and a significant decrease in input energy if the effective power is unchanged. This concentrates the energy of the arc and reduces the total energy fed into the weld pool, which further reduces heat input to the sides of the welding track. As a result, the heat-affected zone in the base metal becomes narrow. Hence, the welding speed negatively correlates with the input heat. It should be noted that the heat-affected zone significantly affects the mechanical properties of the joint, and a large heat-affected zone could lead to joint failure or fractures. The heat-affected zones were analyzed by carrying out hardness tests on the welded joint. The results of these tests are shown in Figure 19.

DP-MIG welding and TP-MIG welding were performed at a constant current of 88 A, and the welding speed was gradually increased from 0.6 m/min to 1.0 m/min. The resulting joints are shown in Figures 20 and 21. The Vickers hardnesses of the welded joints formed at 0.6 m/min and 1.0 m/min were determined with a load of 0.98 N and loading time of 15 s. Hardness tests were first conducted on the base metal to perform detailed analysis of changes in the welded joint hardness; it was found that the hardness of the base metal was 38 HV. The hardness of each zone of the weld is shown in Figure 22.

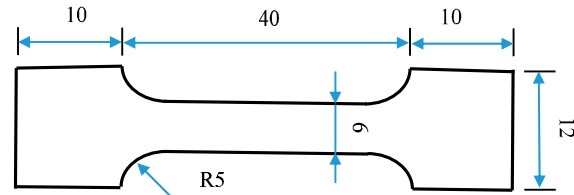

**Figure 19.** Sample extraction of welded joint.

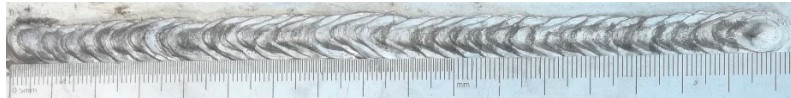

**Figure 20.** DP-MIGW welding joint at 1.0 m/min.

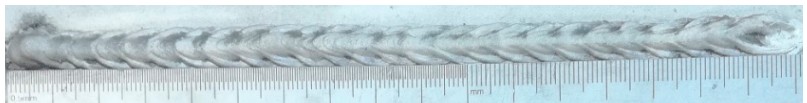

**Figure 21.** TP-MIGW welding joint at 1.0 m/min.

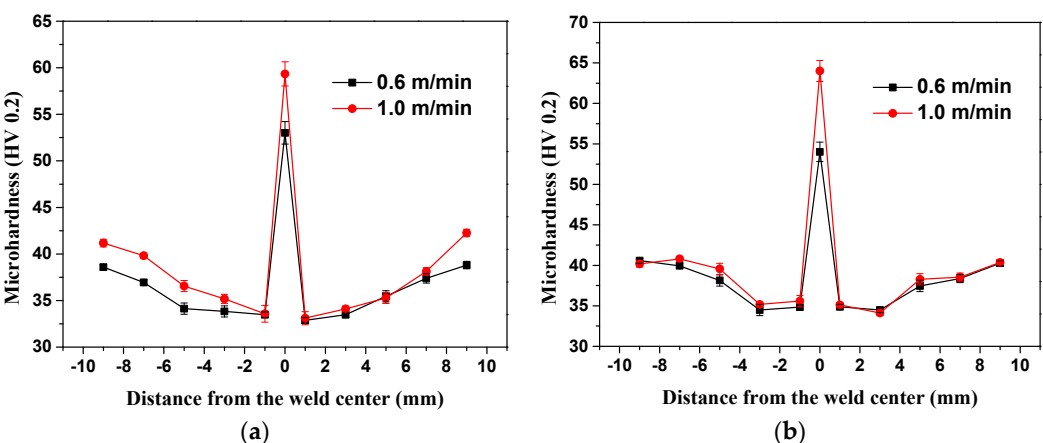

**Figure 22.** Hardness distribution of welded joints at two speeds: (**a**) Comparison of DP-MIGW hardness of welded joints; (**b**) Comparison of TP-MIGW hardness of welded joints.

The hardness of the weld at its center is closely related to that of the welding wire. Figure 23 indicates that welded joints formed at 1.0 m/min are generally stronger than joints formed at 0.6 m/min. In DP-MIG welding, an increase in the welding speed increases the hardness by 19% and decreases the area that is softer than the base metal (i.e., softened zone) by 21%. In TP-MIG welding, an increase in the welding speed increases the hardness by an average of 17% and decreases the softened zone by 27%. The change in hardness with welding speed is shown in Figure 23. The figure shows that an appropriate increase in welding speed helps control energy input in thin-plate welding and is beneficial for reducing the spread of the heat-affected zone. By comparing the results of DP-MIG welding and TP-MIG welding, it was found that the latter approach resulted in harder joints. In addition, it was shown that the input energy can be reduced in both approaches. However, the heat-affected zone in TP-MIG welding was slightly smaller, and the decrease in input energy due to the increase in the welding speed was more pronounced in TP-MIG welding.

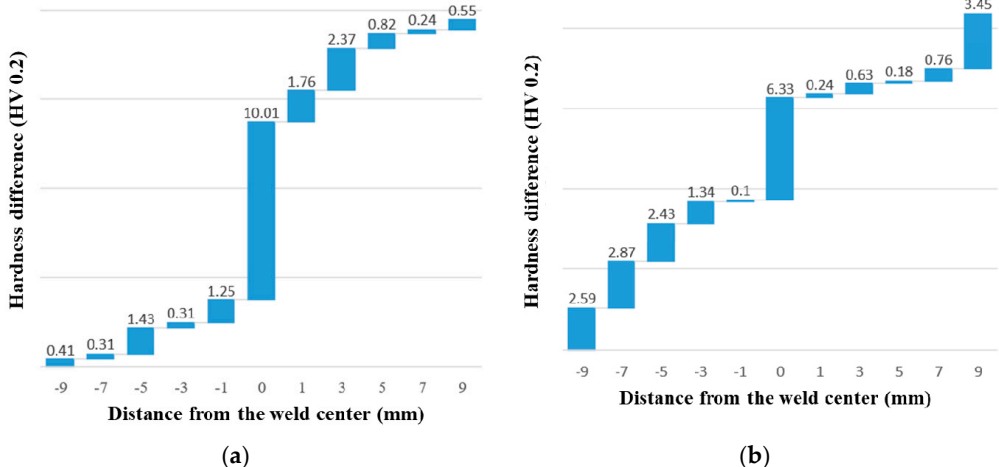

(**a**)    (**b**)

**Figure 23.** Difference in the hardness of welded joints at two speeds: (**a**) DP-MIGW hardness of welded joint; (**b**) DP-MIGW hardness of welded joint.

Figure 24 shows the microstructure of welded joints welded by DP-MIG and TP-MIG at welding speeds of 0.6 m/min and 1.0 m/min. It is observed that the grain size of the fusion zone and the heat-affected zone at the welding speed of 1.0 m/min is less coarsening than that at the welding speed of 0.6 m/min. The reason for this phenomenon may be that the welded joints obtained at the welding speed of 1.0 m/min have less heat input. Besides, welded joints of TP-MIG have less pores than that of DP-MIG, which may due to the effect of high- to low-energy transition pulse groups and low-energy transition pulse groups in TP-MIG welding. High- to low-energy transition pulse groups and low-energy transition pulse groups can make the arc state more stable; therefore, less air is trapped into the molten pool.

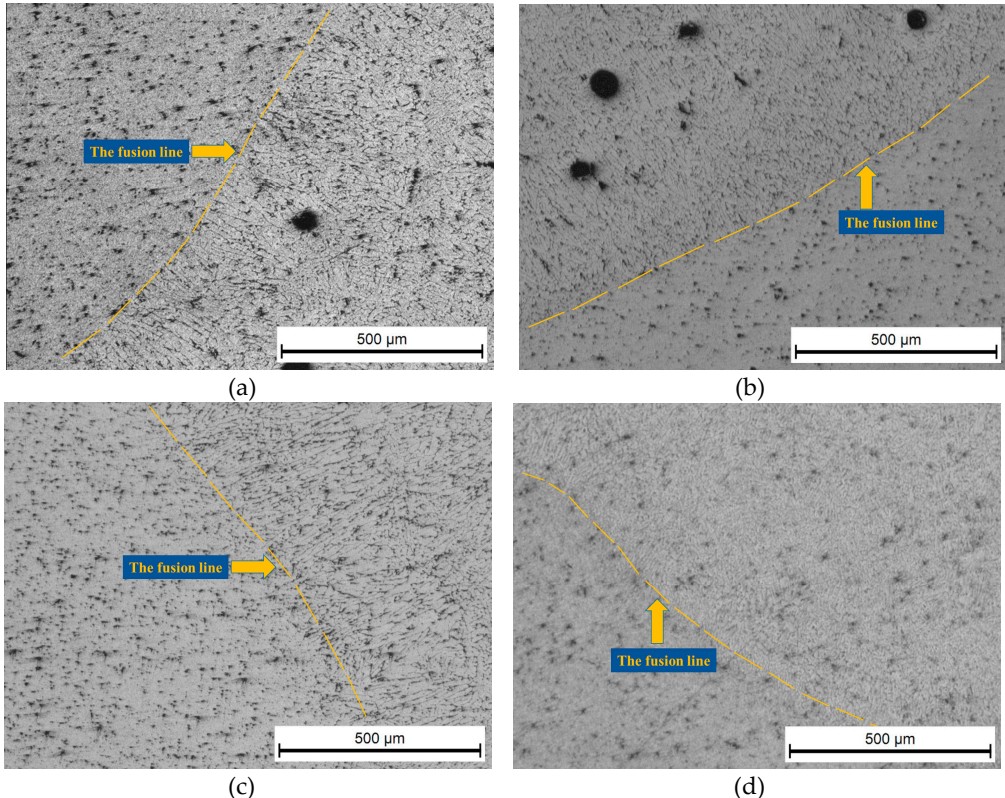

**Figure 24.** Microstructure photograph of welded joints: (**a**) DP-MIGW, 0.6 m/min, (**b**) DP-MIGW, 1.0 m/min, (**c**) TP-MIGW, 0.6 m/min, (**d**) TP-MIGW, 1.0 m/min.

Figure 25 shows the tensile curves and the ultimate tensile strength of the welded joints. The ultimate tensile strength of DP-MIG joints obtained at the welding speeds of 0.6 m/min and 1.0 m/min is 154 MPa and 148 MPa, respectively. The ultimate tensile strength of TP-MIG joints obtained at the welding speeds of 0.6 m/min and 1.0 m/min is 165 MPa and 168 MPa, respectively, which is better than the tensile results of DP-MIG joints.

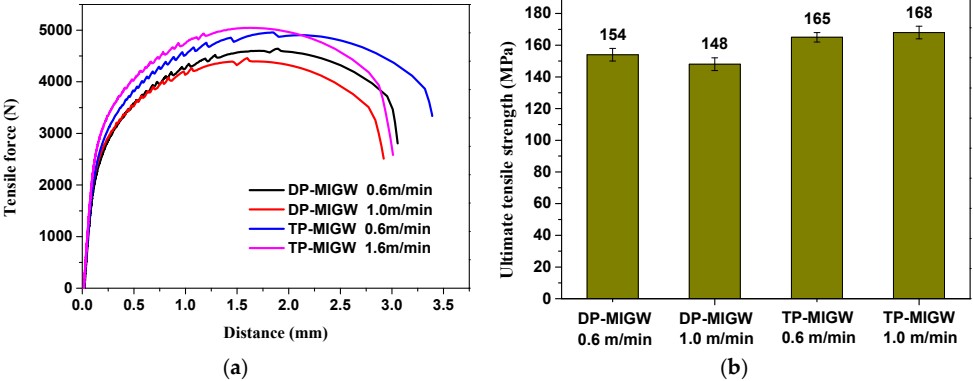

**Figure 25.** Tensile properties of welded joints: (**a**) Tensile curves; (**b**) Ultimate tensile strength.

Figure 26 shows photographs of fractured tensile samples welded by DP-MIG and TP-MIG at the welding speeds of 0.6 m/min and 1.0 m/min. All tensile fractures reveal that there are many cleavage surfaces formed consisting of tearing fibers and steps around dimples, which indicates that the fracture belongs to a plastic-tough mixed fracture mode. In addition, in the fractured tensile samples of DP-MIG, some pores can be clearly seen, which may be the important reason for the low tensile properties of DP-MIG joints.

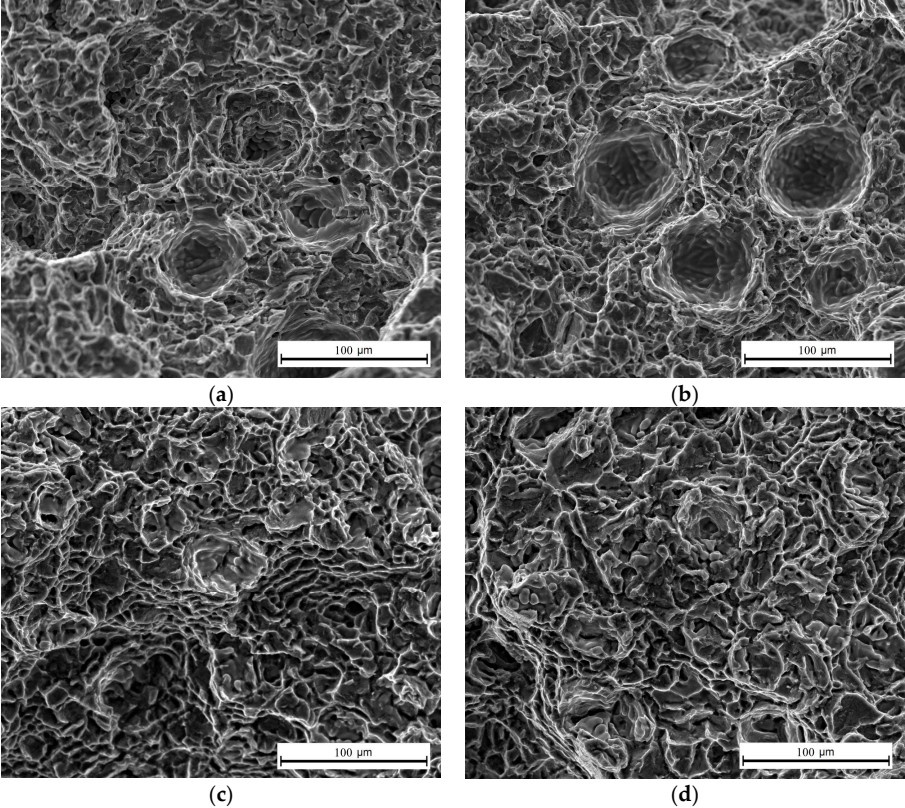

**Figure 26.** Photographs of fractured tensile samples: (**a**) DP-MIGW, 0.6 m/min, (**b**) DP-MIGW, 1.0 m/min, (**c**) TP-MIGW, 0.6 m/min, (**d**) TP-MIGW, 1.0 m/min.

## 5. Conclusions

In this work, a comparative analysis was performed on the process stability and heat input controllability of TP-MIG welding and DP-MIG welding. The following conclusions are drawn from the experimental results:

1.  Owing to abrupt changes in current induced by pulse train transitions in DP-MIG welding, the welding arc makes a sharp sound. In addition, the probabilities of welder burn-back and the formation of screw-like wires during long welding operations are higher in DP-MIG welding than in TP-MIG welding. In TP-MIG welding, the presence of transitional pulses in the pulse train transitions helps stabilize the welding process and soften the sound of the welding arc. The weld formed by TP-MIG welding has a high level of aesthetic quality.

2.  The input energy can be reduced by adjusting the number of pulses in the low-energy pulse train or the base time of the pulses. It was found that varying the number of pulses can reduce the current by 8 A and the input energy by approximately 14%. The welds formed by TP-MIG welding have finer grain sizes than those formed by DP-MIG welding. Under the same conditions, the input energy can be reduced by 12% by adjusting the base time of the pulses. In addition, the density of the fish scales increases as the frequency decreases.

3.  The energies produced by TP-MIG welding and DP-MIG welding decrease as the welding speed increases. An increase in the welding speed from 0.6 m/min to 1.0 m/min decreases the area of the softened zone by 21% and 27% in DP-MIG welding and TP-MIG welding, respectively. In addition, the joint formed by TP-MIG welding is harder than the joint formed by DP-MIG welding. Additionally, the joint formed by TP-MIG welding has better tensile properties than the joint formed by DP-MIG welding. Therefore, TP-MIG welding is a superior technique in terms of its efficacy in reducing welding energy input.

**Author Contributions:** M.X. and X.J. jointly conceived and designed the experiment. M.X., Z.Z. and J.L. jointly performed the experiment and conducted data analysis. M.X. and W.W. analyzed the data and plotted the figures. M.X. and X.J. wrote this paper. Z.Z. and W.W. proofread and translated the paper. X.J. revised the paper, supervised research, and provided financial support.

**Funding:** This paper was supported by the National Natural Science Foundation project of China (51875213) and the High-Level Leading Talent Introduction Program of Guangdong academy of sciences [grant numbers 2016-GDASRC-0106]; Natural Science Foundation of Fujian [grant numbers 2018J01503].

**Acknowledgments:** 

**Conflicts of Interest:** The authors declare no conflict of interest.

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
