# Peer review of "Stability and Heat Input Controllability of Two Different Modulations for Double-Pulse MIG Welding"

_applsci, doi:10.3390/app9010127_

Round 1

Reviewer 1 Report

This is a very interesting work that fits well within the scope of this Journal. However, to my opinion, some aspects need to be addressed prior to publication of this article. The paper is rejected.

In this work, the authors investigated the welding stability and heat input controllability of two methods for the switching of strong and weak pulses, i.e., rectangular-pulse MIG (DP-MIG) welding and trapezoidal-pulse MIG (TP-MIG) welding. The paper is well organised and the authors give just more information about the aspect of weld seem obtained with different welding parameters. They did not discuss about the mechanical properties of joints obtained with different welding parameters. Moreover the welding input has a high influence on temperature distributions in the joint and then on the residual stresses distributions. In my opinion the authors should improve the paper considering this aspect. They should show the different temperature distributions in the joint when the welding parameters are changed. They should carry out other experimental tests e.g.: mechanical testing to obtain the properties of joints (yield strength, ultimate strength, etc.), measure the temperature in the joints, etc. The introduction does not provide sufficient background. Some improvements to the literature reference should be made. Some interesting works can be considered... for example:

Gery, D., Long, H., Maropoulos, P. Effects of welding speed, energy input and heat source distribution on temperature variations in butt joint welding. Journal of Materials Processing Technology. 2005; 167 (2-3): 393-401.

Armentani, E., Esposito, R., Sepe, R. The influence of thermal properties and preheating on residual stresses in welding. International Journal of Computational Materials Science and Surface Engineering. 2007; 1 (2): 146-162.

Hazvinloo, H.R., Honarbakhsh Raouf, A. Effect of gas-shielded flux cored arc welding parameters on weld width and tensile properties of weld metal in a low carbon steel. Journal of Applied Sciences. 2010; 10 (8): 658-663.

Armentani, E., Pozzi, A., Sepe, R. Finite-element simulation of temperature fields and residual stresses in butt welded joints and comparison with experimental measurements (2014) ASME 2014 12th Biennial Conference on Engineering Systems Design and Analysis, ESDA 2014.

Somashekara, M.A., Naveenkumar, M., Kumar, A., Viswanath, C., Simhambhatla, S. Investigations into effect of weld-deposition pattern on residual stress evolution for metallic additive manufacturing. International Journal of Advanced Manufacturing Technology. 2017; 90 (5-8): 2009-2025.

Hang, Z., Wu, D., Zou, Y. Effect of bypass coupling on droplet transfer in twin-wire indirect arc welding. Journal of Materials Processing Technology. 2018; 262: 123-130.

Wen, C., Wang, Z., Deng, X., Wang, G., Misra, R.D.K. Effect of Heat Input on the Microstructure and Mechanical Properties of Low Alloy Ultra-High Strength Structural Steel Welded Joint. Steel Research International. 2018; 89 (6): art. no. 1700500.

Reis, R.P., Scotti, A., Norrish, J., Cuiuri, D. Investigation on welding arc interruptions in the presence of magnetic fields: Arc length, torch angle and current pulsing frequency influence. IEEE Transactions on Plasma Science. 2013; 41 (1), art. no. 6384809: 133-139.

Other minor comments:

Line 41: The authors should explain the acronym I.A.C.S.

Line 51: increase the size font of reference [6].

Figures 1 and 2: the authors should join the two figures and recall each image Figure 1a and Figure 1b.

Figure 3: increase the quality and resolution of figure.

Figure 4: increase the quality and resolution of figure.

Figure 5: increase the quality and resolution of figure.

Figure 6: increase the quality and resolution of figure.

Figure 7: increase the quality and resolution of figure.

Figure 8: the figures 8a and 8b have poor resolution and quality. Please increase them. It is impossible to read the numerical values on the axes.

Figures 12 and 14: increase the quality and resolution of figure. It is impossible to read the numerical values on the axes.

Figures 17 and 18: increase the quality and resolution of figure. It is impossible to read the numerical values on the axes.

Figure 23: increase the quality and resolution of figure.

In the text there are many typos and extensive proof of English is due.

Author Response

This is a very interesting work that fits well within the scope of this Journal. However, to my opinion, some aspects need to be addressed prior to publication of this article. The paper is rejected.

1.In this work, the authors investigated the welding stability and heat input controllability of two methods for the switching of strong and weak pulses, i.e., rectangular-pulse MIG (DP-MIG) welding and trapezoidal-pulse MIG (TP-MIG) welding. The paper is well organised and the authors give just more information about the aspect of weld seem obtained with different welding parameters. They did not discuss about the mechanical properties of joints obtained with different welding parameters. Moreover the welding input has a high influence on temperature distributions in the joint and then on the residual stresses distributions. In my opinion the authors should improve the paper considering this aspect. They should show the different temperature distributions in the joint when the welding parameters are changed. They should carry out other experimental tests e.g.: mechanical testing to obtain the properties of joints (yield strength, ultimate strength, etc.), measure the temperature in the joints, etc. The introduction does not provide sufficient background. Some improvements to the literature reference should be made. Some interesting works can be considered... for example:

Gery, D., Long, H., Maropoulos, P. Effects of welding speed, energy input and heat source distribution on temperature variations in butt joint welding. Journal of Materials Processing Technology. 2005; 167 (2-3): 393-401.

Armentani, E., Esposito, R., Sepe, R. The influence of thermal properties and preheating on residual stresses in welding. International Journal of Computational Materials Science and Surface Engineering. 2007; 1 (2): 146-162.

Hazvinloo, H.R., Honarbakhsh Raouf, A. Effect of gas-shielded flux cored arc welding parameters on weld width and tensile properties of weld metal in a low carbon steel. Journal of Applied Sciences. 2010; 10 (8): 658-663.

Armentani, E., Pozzi, A., Sepe, R. Finite-element simulation of temperature fields and residual stresses in butt welded joints and comparison with experimental measurements (2014) ASME 2014 12th Biennial Conference on Engineering Systems Design and Analysis, ESDA 2014.

Somashekara, M.A., Naveenkumar, M., Kumar, A., Viswanath, C., Simhambhatla, S. Investigations into effect of weld-deposition pattern on residual stress evolution for metallic additive manufacturing. International Journal of Advanced Manufacturing Technology. 2017; 90 (5-8): 2009-2025.

Hang, Z., Wu, D., Zou, Y. Effect of bypass coupling on droplet transfer in twin-wire indirect arc welding. Journal of Materials Processing Technology. 2018; 262: 123-130.

Wen, C., Wang, Z., Deng, X., Wang, G., Misra, R.D.K. Effect of Heat Input on the Microstructure and Mechanical Properties of Low Alloy Ultra-High Strength Structural Steel Welded Joint. Steel Research International. 2018; 89 (6): art. no. 1700500.

Reis, R.P., Scotti, A., Norrish, J., Cuiuri, D. Investigation on welding arc interruptions in the presence of magnetic fields: Arc length, torch angle and current pulsing frequency influence. IEEE Transactions on Plasma Science. 2013; 41 (1), art. no. 6384809: 133-139.

Response 1: Thank you for your kind comments and useful suggestions! We have added Some literature references [6-13] in the introduction (lines 39-42 and 44-45).

Other minor comments:

2.Line 41: The authors should explain the acronym I.A.C.S.

Response 2: Thank you for your useful suggestions!  I.A.C.S. is the acronym of "International Annealed Copper Standard" (line 40).

3.Line 51: increase the size font of reference [6].

Response 3:  Thanks for your careful checking. We have increased the size font of reference [14] (line 52).

4.Figures 1 and 2: the authors should join the two figures and recall each image Figure 1a and Figure 1b.

Response 4: Thank you for your useful suggestions! Figure 1 displays schematically the relationship between transfer mode and current, while Figure 2 demonstrates the pulsed current. There is not a strong connection between them, hence they are kept as independent figures. However, we still appreciate greatly your kind suggestions!

5.Figure 3: increase the quality and resolution of figure.

Response 5: Thank you for your useful suggestions! We have changed a higher quality and resolution image of Figure 3.

6.Figure 4: increase the quality and resolution of figure.

Response 6: Thank you for your useful suggestions! We have changed a higher quality and resolution image of Figure 4.

7.Figure 5: increase the quality and resolution of figure.

Response 7: Thank you for your useful suggestions! We have changed a higher quality and resolution image of Figure 5.

8.Figure 6: increase the quality and resolution of figure.

Response 8: Thank you for your useful suggestions! We have changed a higher quality and resolution image of Figure 6.

9.Figure 7: increase the quality and resolution of figure.

Response 9: Thank you for your useful suggestions! We have changed a higher quality and resolution image of Figure 7.

10.Figure 8: the figures 8a and 8b have poor resolution and quality. Please increase them. It is impossible to read the numerical values on the axes.

Response 10: Thank you for your useful suggestions! We have changed a higher quality and resolution image of Figure 8.

11.Figures 12 and 14: increase the quality and resolution of figure. It is impossible to read the numerical values on the axes.

Response 11: Thank you for your useful suggestions! We have changed a higher quality and resolution image of Figures 12 and 14.

12.Figures 17 and 18: increase the quality and resolution of figure. It is impossible to read the numerical values on the axes.

Response 12: Thank you for your useful suggestions! We have changed a higher quality and resolution image of Figures 17 and 18.

13.Figure 23: increase the quality and resolution of figure.

In the text there are many typos and extensive proof of English is due.

Response 13: Thank you for your useful suggestions! We have changed a higher quality and resolution image of Figure 23.

Reviewer 2 Report

Can you describe the model or this 2 modulation method limitation? 

Author Response

Can you describe the model or this 2 modulation method limitation? 

Response: Thank you for your useful suggestions! The model has been described in fig.3 and fig.4 and in lines 101-109 and 130-138.

Reviewer 3 Report

The paper entitled "Stability and heat input controllability of two different modulations for double-pulse MIG welding” by Xue et al. deals with the comparison of the stability and het input controllability during TP-MIG and DP-MIG welding. The structure of the modulation and the experimental comparison of both approaches, during welding of aluminium is studied. The final appearance of the welds, microstructure, and hardness is compared.

Results are in the scope of the Applied Sciences journal. The findings are interesting, but after reading the paper, I have some comments about it:

GENERAL COMMENTS:

1)     This paper is well-written and the results are interesting; however, I find that authors have include a short study on the porosity formation. As discussed in the text, this is a very important issue in aluminium welding. From the micrographs given in the manuscript, porosity formation cannot be ascertained.

PARTICULAR COMMENTS

1)      Please, improve the quality of the Figs.: 1-7, 8a-b, 12a-b, 14a-b, 17-18. Text in these figures is very difficult to read.

2)     (Page 3) n and m are not defined in Fig. 3.

3)     (Page 7, Line 227) Please, replace “Newton’s interpolation method” with “Newton-Raphson method”

4)     (Page 7, Fig. 7) Y-axis title is absent. Please, correct it. Add also error bars.

5)     (Page 9, Table 1) In section 3, it is stated that welding experiments were done on pure aluminium, but in section 4, it is stated that AA6061 alloy was used. Were two different base materials used in this study?

6)     (Page 9, Lines 274-276) Is U the average voltage? In Table 2 this parameter is designed as V. Please, use only one letter for this parameter. Give a proper reference for the value of the effective power (0.7-0.8) of the energy sources. Why was P_lambda taken as 0.7?

7)     (Page 11, Figs-15-16) Figure captions should be corrected. These figures depict the micrograph of the welds; they are not a phase diagram; “phase gold phase diagram” should be corrected.

8)     (Page 14) Please, add error bars to the hardness values provided in Fig. 22.

Author Response

The paper entitled "Stability and heat input controllability of two different modulations for double-pulse MIG welding” by Xue et al. deals with the comparison of the stability and het input controllability during TP-MIG and DP-MIG welding. The structure of the modulation and the experimental comparison of both approaches, during welding of aluminium is studied. The final appearance of the welds, microstructure, and hardness is compared.

Results are in the scope of the Applied Sciences journal. The findings are interesting, but after reading the paper, I have some comments about it:

GENERAL COMMENTS:

1) This paper is well-written and the results are interesting; however, I find that authors have include a short study on the porosity formation. As discussed in the text, this is a very important issue in aluminium welding. From the micrographs given in the manuscript, porosity formation cannot be ascertained.

 Response 1: Thank you for your kind comments and useful suggestions! As discussed in the text, the DP-GMAW method has the ability to reduce the porosity, hence it is hard to find porosity formation.

PARTICULAR COMMENTS

1)    Please, improve the quality of the Figs.: 1-7, 8a-b, 12a-b, 14a-b, 17-18. Text in these figures is very difficult to read.

Response 1: Thank you for your useful suggestions! We have changed higher quality and resolution images of Figs.: 1-7, 8a-b, 12a-b, 14a-b, 17-18.

2)    (Page 3) n and m are not defined in Fig. 3.

Response 2: Thank you for your useful suggestions! We have defined n and m in Fig. 3 (lines 108-109).

3)    (Page 7, Line 227) Please, replace “Newton’s interpolation method” with “Newton-Raphson method”

Response 3: Thank you for your kind suggestions! We have replaced “Newton’s interpolation method” with “Newton-Raphson method” (line 231).

4)    (Page 7, Fig. 7) Y-axis title is absent. Please, correct it. Add also error bars.

Response 4: Thank you for your kind suggestions! We have added the Y-axis title and it has been added, as you can see in Fig. 7.

5)    (Page 9, Table 1) In section 3, it is stated that welding experiments were done on pure aluminium, but in section 4, it is stated that AA6061 alloy was used. Were two different base materials used in this study?

Response 5: Thank you for your careful checking! We made a mistake of pure aluminum in section 3, and have changed it to AA6061 alloy (line 219).

6)    (Page 9, Lines 274-276) Is U the average voltage? In Table 2 this parameter is designed as V. Please, use only one letter for this parameter. Give a proper reference for the value of the effective power (0.7-0.8) of the energy sources. Why was P_lambda taken as 0.7?

Response 6: Thank you for your kind comments and useful suggestions! U is the average voltage (Page 9, Lines 274-276). In Table 2, V is the unit of parameter U, we wrongly wrote it in italics, which caused your misunderstanding, and we have corrected it.

In the book "Basic Technology and Application of Gas Shielded Welding (in Chinese)", the authors give the Lambda as 0.7-0.8. In our case, more heat is conducted to the air due to the lower circumstance temperature, then the Lambda was taken as 0.7.

7)    (Page 11, Figs-15-16) Figure captions should be corrected. These figures depict the micrograph of the welds; they are not a phase diagram; “phase gold phase diagram” should be corrected.

Response 7: Thank you for your kind suggestions! We have corrected the captions of Figs-15-16 by changing "phase gold phase diagram" to "micrograph of the welds".

8)    (Page 14) Please, add error bars to the hardness values provided in Fig. 22.

Response 8: Thank you for your useful suggestions! We have added error bars to the hardness values provided in Fig. 22 (Page 15).

Round 2

Reviewer 1 Report

The authors have improved highly the paper, but many lacks are still present, major revisions are due. In the following there are my comments.

1) The authors did not discuss about the mechanical properties of joints obtained with different welding parameters. Why? The authors should show the mechanical properties of two joints and compare them and discuss the difference.

2) The different welding parameters influence the welding heat input and so, the distortions and the residual stresses in the joints. The authors should show the temperature distributions of the two joints and discuss .them.

3) Figure 2: the figure is black please correct it.

4) Figure 3: the figure is black please correct it.

5) Figure 19: the figure is black please correct it.

Extensive proof of English is due.

Author Response

Thank you very much for your letter and comments/suggestions concerning our manuscript entitled “Stability and heat input controllability of two different modulations for double-pulse MIG welding” (ID: applsci-391606). The comments and suggestions are very valuable and helpful for us to improve the quality of the paper. We have carefully considered the comments/suggestions and have made a substantial revision to the manuscript. Revised places are marked red in the text of the manuscript. Meanwhile, we have also prepared a list of changes and responses to the review comments as follows:

Reviewer #1

A) The authors did not discuss about the mechanical properties of joints obtained with different welding parameters. Why? The authors should show the mechanical properties of two joints and compare them and discuss the difference.

Response: Thanks for this useful suggestion. The microstructure and tensile properties of joints obtained with DP-MIG and TP-MIG at the welding speeds of 0.6 m/min and 1.0 m/min have been added in the revised manuscript. The relevant test results have been analyzed (page 17~18, line 389-425).

B) The different welding parameters influence the welding heat input and so, the distortions and the residual stresses in the joints. The authors should show the temperature distributions of the two joints and discuss them.

Response: Thanks for your careful checking. The temperature distributions of the two joints have an important effect on microstructures and mechanical properties of welded joints. However, measuring the temperature of the molten pool needs a precise high temperature measurement system that our research group does not have. So, we cannot finish these studies in the near future.

C) Figure 2: the figure is black please correct it.

Response: Thanks for your careful checking. Black image of Fig. 2 may be a display problem, so we have uploaded PDF document format.

D) Figure 3: the figure is black please correct it.

Response: Thanks for your careful checking. Black image of Fig. 3 may be a display problem, so we have uploaded PDF document format.

E) Figure 19: the figure is black please correct it.

Response: Thanks for your careful checking. Black image of Fig.19 may be a display problem, so we have uploaded PDF document format.

F) Extensive proof of English is due.

Response: Thanks for this useful suggestion. We have Polish the manuscript by professional organization, and uploaded the proof of English.

Finally, we appreciate very much for your time in editing our manuscript and the referees for their valuable suggestions and comments. We are looking forward to hearing from your final decision when it is made.

Best wishes!

Jiaxiang Xue, Min Xu, Wenjin Huang, Zhanhui Zhang, Wei Wu, and Li Jin

Round 3

Reviewer 1 Report

Accepted in present form